# Low-Grade Inflammation and Role of Anti-Inflammatory Diet in Childhood Obesity

**DOI:** 10.3390/ijerph20031682

**Published:** 2023-01-17

**Authors:** Ewelina Polak-Szczybyło

**Affiliations:** Department of Dietetics, Institute of Health Sciences, College for Medical Sciences, University of Rzeszow, 35-310 Rzeszow, Poland; ewpolak@ur.edu.pl

**Keywords:** low-grade inflammation, childhood obesity, anti-inflammatory diet, children, metabolic syndrome

## Abstract

At present, pediatric obesity is a significant public health concern. We have seen a surge of disorders that are obesity-related, e.g., insulin resistance (IR), type 2 diabetes mellitus (T2DM), high blood pressure, heart disease, stroke, cancer, non-alcoholic fatty liver disease (NAFLD), autoimmune disorders and many more. The mechanisms linking these diseases to excess body weight are related to low-grade inflammation (LGI). Although there is a limited number of studies assessing this immune process in childhood obesity, they indicate its significant importance for the health of future generations. There is a need for more research into the prevention and treatment of low-grade inflammation in childhood. The aim of this review is to present and discuss the process of “cold” inflammation, and its impact on health and anti-inflammation nutrition. A diet rich in anti-inflammatory nutrients may be the key to maintaining health, as well as recovery.

## 1. Introduction

Obesity is an increasing problem in both children and adults. The World Health Organization (WHO) reports that in 2016 over 340 million children and adolescents aged 5–19 were overweight or obese. Body weight above the norm among children and adolescents aged 5–19 has risen from 4% in 1975 to over 18% in 2016. According to the latest WHO data, in 2019, an estimated 38.2 million children under the age of 5 were overweight or obese [1]. In 2020, during the COVID-19 pandemic, individual countries recorded another increase in weight among children and adolescents. This is the result of isolation, which is conducive to an increase in the number of meals, especially fried foods and sweets, and less exercise and access to healthy food. Additionally, 41.7% of adolescents reported weight gain in Palestine and 25% declared a weight increase in a Spain. The prevalence of obesity increased by more than 15% in many US states and from 10.5% to 12.9% in China [2]. Overweight and obesity are now on the rise in low- and middle-income countries, particularly in urban settings. Globally, overweight and obesity are linked to more deaths than underweight [1]. Excess body weight applies both to mental and physical health. Obesity also affects rising health care system costs, as there are many obesity-related illnesses, e.g., insulin resistance, type 2 diabetes mellitus, cardiovascular disease, autoimmune disorders and cancer [3,4]. The cause of obesity in children is similar to that of adults and obese children express the same comorbidities that are associated with being overweight and obese as adults [5]. The immune system develops throughout childhood with many stages of maturation. Obesity may interfere with this process. The immune processes present in low-grade inflammation in childhood obesity should be carefully investigated in order to identify the early mechanisms leading to the development of some complications and allow optimal intervention.

## 2. Methods

This review is narrative, and no systematic search of the literature was performed. It presents several observational and interventional studies related to childhood obesity, low grade inflammation and diet. Databases such as PubMed/Medline, Scopus and Google Scholar were searched comprehensively. The search time was September 2022. The following keywords in English were used to search for papers: low-grade inflammation, pediatric obesity, childhood obesity, anti-inflammatory diet, anti-inflammatory nutrition, cardiovascular disease, insulin resistance, metabolic syndrome, asthma, obstructive sleep apnea. It was aimed at identifying the most clinically relevant articles related to the topic, rather than to provide a comprehensive overview. A search was limited to English literature. The mechanism of low-grade inflammation is not fully understood. It is assumed that in children it is similar to that in adults. Due to shortcomings in the literature, mechanism of low-grade inflammation in Chapter 3 is described on the basis of data in adults. All studies in Chapter 4, and those related to the dietary patterns, foods and nutrients quoted in Chapter 5, apply to the research on children and adolescents.

## 3. Mechanism of Low-Grade Inflammation in Pediatric Obesity

The details of how adipose tissue effect LGI have not been fully investigated. The mechanism involves adipose tissue in inflammatory responses by several pathways [6]. The greater the amount of adipose tissue, the higher the level of pro-inflammatory markers. The inflammatory role of adipocytes is related to its expansion, hyperplasia and hypertrophy [7]. Cellular stresses such as endoplasmic reticulum (ER) stress, mitochondrial dysfunction and oxidative stress are presented in adipose tissue. The meta-inflammation, excessive ectopic fat accumulation or rising blood glucose levels induced ER stress. Excessive dietary saturated fatty acids or fructose exhibit an impact on ER stress. The factors mentioned above can cause endoplasmic reticulum dysfunction, provoking the accumulation of immature proteins, which triggers the unfolded protein reaction (UPR) in an attempt to restore ER homeostasis. Adipocytes are subject to ER stress due to the high accumulation and metabolism of lipids and some of them undergo apoptosis. Tissue-embedded macrophages capture fat droplets and also engulf dead adipocytes through phagocytes. This process causes a release of a large amounts of reactive oxygen species (ROS), further aggravating mitochondrial dysfunction, ER stress and cytoplasmic pH shift. It results in a change in adipocyte phenotype, leading to increased secretion of adipokines. These factors induce cellular stress, which change adipocyte phenotype and increase adipokine secretion and the severity of inflammation. ER stress also leads to autophagy miss-folded/unfolded proteins in adipocytes [8,9].

LGI is related to the secretion of cytokines and adipokines by adipose tissue. This mechanism is also mediated by the hormone leptin [10]. The second mechanism is related to the hypertrophy and apoptosis of adipocytes. It causes inflammation and increased production of pro-inflammatory adipokines as well a release of free fatty acids with activation of the immune system [11]. Low-grade inflammation is related to the recruitment of neutrophils and pro-inflammatory T cells (Th1 cells), which secrete pro-inflammatory cytokines and stimulate differentiation of monocytes into the proinflammatory macrophage subtype [6]. The macrophages then migrate to the adipose tissue, taking an active part in low-grade inflammation. Resistin produced by adipocytes and white blood cells (WBCs) is involved in the differentiation of pro-inflammatory macrophages (M1), but its exact role has not been discovered so far [11]. Adipocytes also produce IL-6, which increases the secretion of C-reactive protein (CRP) from the liver, and it is the high-sensitive C-reactive protein (hs-CRP) that is extensively studied in pediatric obesity [1]. It often determinates the cardiometabolic and metabolic syndrome risk in adults [12], which is associated with low-grade inflammation and increased level of inflammatory markers, leading to a negative effect on health. The severity of LGI is related to the amount of adipose tissue. Inflammatory markers levels are highest among obese children when compared to normal and overweight children. The risk of metabolic syndrome is higher with worsening obesity [13,14]. In children as young as 3 years of age, there occurs a strong relation between waist circumference or body mass index and the level of CRP, IL-6 and tumor necrosis factor α (TNF-α) [1]. A simplified mechanism of low-grade inflammation is shown in Figure 1.

## 4. Impact of Childhood Obesity and Low-Grade Inflammation on Health

Childhood and adolescent overweight or obesity significantly increases the risk of civilization disorders causing a decreased quality of life, lower productivity and disabilities [15]. The possible consequences of overweight among children and adolescents associated with LGI are presented in Figure 2.

Obese children have the highest level of inflammatory markers and the highest risk of metabolic syndrome compared to those who are overweight or of normal weight. In an observational study in 439 obese, 31 overweight and 20 normal-weight children, it was concluded that risk increases in accordance with the severity of obesity [16]. Although LGI is an important element in the pathophysiology of some cardiometabolic diseases, the mechanism linking them has not been sufficiently elucidated yet. Markers have been shown in LGI damage of the innermost layers (intima media) and lining [endothelium) of blood vessels. It causes atherosclerotic lesions, hypertension and thrombosis [17,18,19]. That damage can be observed in children even in cases of mild obesity [20]. In the research by Baba et al., 1871 boys and 1810 girls aged 12 were examined. When it comes to both girls and boys, the excess body weight classified as obesity was associated with greater systolic blood pressure. In contrast, diastolic blood pressure was related with boys’ body composition, both in obese and overweight patients [21]. Serum triglycerides (TG) levels were elevated in both boys and girls who had excess body fat. It should also be noted that high levels of TG appeared in girls when they were obese, and in boys when they were overweight. Reduced HDL levels were correlated with the body weight composition of both overweight and obese children, no matter their sex. Elevated levels of white blood cells (WBCs) and platelet count were correlated with excess body fat in both sexes as well in the case of excess weight [21]. As reported by Morten et al., the risk of cardiovascular disease increases in childhood obesity, but is not related to the degree of adiposity. In addition, it was noted that the risk varies depending on gender. Hs-CRP and WBC indicators are said to be the most strongly associated with different cardiometabolic risks and comorbidities. However, determining the level of risk is very difficult and depends on individual characteristics [9].

In the childhood and pubertal period, there is a physiological increase in insulin resistance and high insulin levels which usually normalize with the end of puberty [22]. However, in pediatric obesity, the inflammatory markers negatively influence insulin sensitivity and glucose transport [17,23]. Children with excess body weight have lower insulin sensitivity than children with normal body weight [24]. Distribution of body fat is the most important. Visceral adipose tissue has the greatest impact on blood levels of adiponectin, CRP, interleukin-6 and the degree of systemic inflammation in adolescents [25,26]. In research, elevated levels of CRP, TNF-α and IL-6 were associated with the excess body weight and a higher risk of IR [13]. TNF-α and IL-6 indirectly inhibit the activation of insulin signaling and sustaining insulin resistance. Hyperglycemia affects the increased production of IL-6 by macrophages, which in turn intensifies the already existing IR [27]. Insulin resistance is an intermediate stage in the natural progression to type 2 diabetes mellitus [22].

With the growing number of obese children, non-alcoholic fatty liver disease has become the most common pediatric liver disease. According to research, this problem may affect up to 38% of obese children [28]. In studies on over 3000 children, the level of aminotransferases (AST and ALT) was correlated with the content of adipose tissue. ALT levels were higher in obese and overweight boys and girls, and AST was elevated in obese boys only [21]. The pathogenesis of NAFLD in children and adults is the same and multifactorial. LGI is not the only risk factor, but fatty liver associated with obesity promotes inflammation in the liver [29].

Childhood asthma has been shown to be associated with obesity, often without an underlying allergy. Both obesity and asthma have shown disturbed levels of leptin, adiponectin, resistin and visfatin [30]. As reported by Magron et al., children from three groups—with obesity, with asthma and with obesity and asthma—were examined for the selected cytokines and adipokines. There was also a control group. Children with obesity and asthma had the highest levels of leptin, IL-2 and INF-γ, as well as the lowest level of adiponectin compared to the other groups. Other cytokines (IL-1β, IL-6, IL-8, IL-13) did not show this tendency. The same conclusions were drawn from the research by Krastev et al. The level of IL-6, CRP, TNF-α was determined in 88 children. Obesity and asthma enhance the pro-inflammatory state in the body because high levels of leptin stimulate Th1 response, and low levels of adiponectin inhibit IL-10 secretion [31]. In research by Castro-Rodríguez et al., BMI positively correlates with the prevalence of asthma in boys and girls. Girls between 6 and 11 who were overweight or obese were seven times more likely to have asthma at the age of 11 to 13 [32]. Consequently, weight reduction helps to improve pulmonary functions and asthma symptoms, as well as reduces the need to use rescue bronchodilators and the frequency of asthma exacerbations [33]. Asthma can be a consequence of the changes in gut microbiome and short-chain fatty acid (SFA) circulation which can lead to some allergic airway diseases. Reduced intake of fiber in obese patients combined with rich fat intake may lead to a reduction in bacterial colonies and SFA levels in people with obesity and asthma [34,35].

It is suspected that changes in the microbiome in obesity can also influence the risk of depression [36]. Obesity and depression are undeniably linked. Depression in childhood often results in obesity in adulthood, and obesity in adolescence is associated with an increased risk of depression in adulthood [37]. Miller et al. concluded that when it comes to young adults, the symptoms of depression contribute to increased obesity and, consequently, to inflammation. Children aged 14 with a higher BMI who were examined at the age of 17 more often suffered from excess body weight, inflammation, depression and other mental health problems [38].

Obesity is often a risk factor for the obstructive sleep apnea syndrome (OSAS). In both cases, these diseases are inflammatory of a low intensity. Patients with OSAS have been shown to have higher levels of IL-6. The mechanism of the inflammatory response linking OSAS to LGI has not been fully understood yet [39].

Obesity is also a risk factor for the development of idiopathic intracranial hypertension (IIH). This mechanism has not been fully explained, but it is believed that the compound is involved in pro-inflammatory markers secreted by adipocytes. It has also been found that it may be related to impaired levels of leptin, TNF-α, IL-1, IL-6 and, indirectly, cortisol [22].

The review by Stolzman et al. has shown that increased inflammatory markers in obese children lead to the development of risk factors for diseases which are typical of adulthood [13]. Obese children often grow up to be obese adults. Moreover, long exposure to low-grade inflammation may increase the risk of morbidity [13].

## 5. The Diet and Lifestyle Changes in Childhood Obesity

In general, obesity is caused by poor nutrition. Modern man’s diet contains a significant amount of pro-inflammatory components that enhance the existing low-grade inflammation, [40]. A total of 286 healthy children aged 11 years, with different body mass, had complete dietary information, anthropometric indicators and serum concentrations of hs-CRP, IL-6, adiponectin and leptin. In this observational study, it was proven that a Western-type diet with excess calories, ultra-processed food, high in red meat, refined grains, high-fat dairy, saturated or trans-fatty acids and low in omega-3 polyunsaturated fatty acid (PUFA), contributes to the increased secretion of inflammatory markers [41]. According to the observational research by Oddy et al., Western-type diet in adolescents lead to an increased level of CRP and IL-6 in healthy males and females [42]. These conclusions were confirmed by cross-sectional studies among 670 Iranian adolescent girls after completing a food frequency questionnaire [43]. Gonzalez-Gil et al. showed that soft drinks with sugar, mayonnaise and milled cereals increased the level of hs-CRP [44]. The research was conducted on 16,228 participants aged 2–10 years [44]. Additionally, high intake of sodium leads to higher levels of TNF-α. It was confirmed in a cross-sectional study involving 766 healthy adolescents (14 to 18 years old). Dietary sodium intake was estimated by seven-day 24 h dietary recall. Participants were measured for leptin, adiponectin, C-reactive protein, tumor necrosis factor-α and intercellular adhesion molecule-1. [45]. In contrast, observational research on the adolescent age group by Del Mar Bibiloni et al. showed that it is not the dietary factor but body mass index and waist-to-height ratio (WHtR) showed a positive correlation with the level of hs-CRP. The mechanisms by which dietary patterns affect the inflammatory process are explored to some extent [46]. The type of diet may induce a pro-inflammatory or anti-inflammatory response by the influence of properties of their constituents [47].

A Mediterranean diet (MD), Dietary Approaches to Stop Hypertension diet (DASH) and a diet with low glycemic index are associated with decreased levels of biomarkers, including CRP, IL-6 and TNF-α [41]. The Mediterranean diet is a combination of anti-inflammatory nutrients, which is why it is often recommended as a model of nutrition promoting health and a slim figure. It consists of whole grains, legumes, nuts, fish and low-fat dairy, healthy oils such as olive oil, vegetables, fruit rich in antioxidants, folate and flavonoids. The Mediterranean diet is characterized by low consumption of meat and processed food. Numerous studies confirm its effectiveness in reducing inflammatory markers. An observational study by Sured et al. on 598 adults and adolescents of both sexes indicates decreasing influence on hs-CRP level [48]. The cross-sectional analysis with 1462 adolescents (625 girls) aged 9–18 showed the same results [49]. It has also been shown that the MD decreases the level of IL-6 and TNF-α [50]. The Mediterranean diet introduced in 44 children with asthma lowers IL-17 levels [51].

Another recommended dietary intervention for lowering inflammation is a diet with a low glycemic index. In research on girls with excess body weight, a diet with a low glycemic index was used for 10 weeks. Compared to the control group, the low-glycemic-index diet successfully lowered hs-CRP and IL-6 [52]. Different research also confirms the beneficial effect of a low-glycemic-index diet on the level of hs-CRP in children [53]. In this research, there were two types of diets used among obese children, the hypocaloric low-glycemic-index diet and the hypocaloric high-glycemic-index diet. The results related to weight loss, blood pressure and hs-CRP levels were similar, while metabolic indicators such as triglycerides were significantly lower in the group on the low-glycemic-index diet. This may indicate that the glycemic index in itself is not beneficial in reducing low-grade inflammation markers. Weight reduction is even more important [54]. Inconclusive results were also shown in the interventional research by Damsgaard et al. on obese children [55].

The Dietary Approaches to Stop Hypertension diet increase the consumption of vegetables and fruits, lean meat and dairy products. In DASH, the amount of sodium is reduced to approximately 1500 mg/day [56]. The randomized, cross-over clinical trial on adolescents shows a significant reduction in high-sensitivity C-reactive protein after the DASH eating pattern for 6 weeks [57].

Not only dietary patterns indicate the anti-inflammatory effect, but also food groups; macro- and micronutrients can reduce low-grade inflammation in obese people. High consumption of fruit, vegetables, fish and whole grains was inversely associated with BMI and inflammation in 17-year-old adolescents [42]. Fruit and/or vegetables have particularly beneficial anti-inflammatory properties. Many cross-sectional studies describe the benefits of consumption of this group of food on LGI. High vegetable and/or fruit intake is associated with lower hs-CRP [42,58] and IL-6 [59,60]. On the other hand, lowering of TNF-α was showed only in healthy females [61]. In patients with asthma, increasing consumption of vegetables and fruit results in lowering of IL-17 [62]. Intake of fruits and nuts was also associated with a decreased level of IL-4 [61]. Another significant group of foods are grains. Their impact on health depends on the degree of their refined processing. The consumption of whole grains decreased the level of hs-CRP among obese children in a randomized crossover clinical trial [63] and in intervention research in adolescents with normal weight [64]. Adolescents who frequently consume cereals and roots had lower levels of IL-6, IL-10 [61]. IL-17 was lower among adolescents with asthma who eat whole grains more often [62]. In the same research, higher dairy intake was associated with higher levels of IL-17F [62]. In addition, the intake of dairy products was positively correlated with the levels of IL-1, IL-5, IL-6, IL-10, TGFβ-1 in 464 adolescents (13–17 years old) [61]. However, most of the cross-sectional studies have not shown any connection between dairy products intake and the level of inflammatory biomarkers [44,60,65,66]. Higher frequency of legume intake showed decreased hs-CRP levels in adolescents and children [60,61]. It should be mentioned that the consumption of certain groups of products increases the intensity of inflammation in the body. The frequent consumption of meat in observational research by Aeberli et al. was associated with elevated levels of IL-6, regardless of the body mass index [67]. A relationship was also found between frequent meat consumption and high levels of IL-2 and IL-10 [61]. Increased sugar-sweetened beverage intake is associated with increased C-reactive protein concentrations [68]. The products mentioned above have pro-inflammatory or anti-inflammatory properties mainly due to the nutrients they contain.

The observational studies in healthy children and adolescents found that saturated fatty acids intake was associated with higher levels of CRP [67,69,70,71]. Monounsaturated fatty acid (MUFA) and omega-3 polyunsaturated fatty acid are anti-inflammatory nutrients [45,61]. In contrast, the overall content of dietary fat, not necessarily its quality, is irrelevant to the levels of inflammatory markers (hs-CRP, IL-6) [72,73]. It emerged that omega-3 PUFA in the diet inhibits activation of pro-inflammatory pathways and reduces cytokine level [74]. In an observational study on 69 children, it was shown that the consumption of antioxidant vitamins (vitamins E and C and β-carotene) was not a predictor of the level of CRP, IL-6 or TNF-α [68]. In a cross-sectional study with 285 adolescent boys and girls aged 13 to 17, serum C-reactive protein level was significantly inversely associated with the intake of vitamin C and folate. Serum interleukin-6 was inversely associated with the intake of beta carotene and vitamin C. Serum tumor necrosis factor-α was inversely associated with beta carotene and luteolin [59]. Other research also confirms association between dietary intake of vitamins, specifically A, C and E, and lower levels of hs-CRP and IL-6 [45,46,59,69]. Several observational researchers [65,69,70,72,75,76,77] and one intervention [78] investigated the effect of fiber on inflammation in children and adolescents, both with normal and with excessive body weight. The results did not conclusively confirm the anti-inflammatory properties of fiber consumption. As reported by Navarro et al. and Miller et al., high fiber intake has been shown to be associated with lower inflammation in the body [69,77]. It should be noted that fiber supports the proper intestinal microbiota, which have a significant impact on the immune processes in the body. Normal intestinal microflora regulates the inhibition or production of pro-inflammatory chemokines and cytokines [79]. A pro-inflammatory and anti-inflammatory effect of nutritional factors on the level of markers is shown in Table 1.

## 6. Limitations

Obviously, this review is not free from limitations. The first and the most important one is the lack of articles describing the mechanism of LGI in childhood obesity. Current research is usually observational and is influenced by environment, lifestyle, genes and many other factors beyond diet and body weight. In addition, inclusion and exclusion criteria are often poorly defined in such studies. The differences between LGI in children, adolescents and adults are not defined. Another limitation is changes in the levels of inflammatory markers, resulting primarily from a decrease in body weight or body composition and not directly from diet. Nutrition in that case can only play an additional aspect. Additionally, energy restriction may explain these changes, not the properties of diet patterns, foods groups or nutrients. The influence of genetics, gender or ethnicity on LGI and response to diet should also be more investigated. Studies in children are limited. For this reason, there is a great need for further research to prevent inflammatory diseases.

## 7. Conclusions

Obese children are at high risk of low-grade inflammation which can affect their lives, as obese children are likely to become obese adults. This prolonged exposure to inflammatory markers may increase the risk of developing many diseases at an early age. An increased BMI in childhood is a predictor of the occurrence of metabolic syndrome in adulthood. Among obese children, the intensity of low-grade inflammation should be identified for the purpose of prediction of diseases such as cardiovascular disease, T2DM, etc. It will also allow for the development of an appropriate dietary intervention reducing excessive adiposity and involving anti-inflammatory components of the diet to mitigate some negative health consequences in adulthood.

## Figures and Tables

**Figure 1 ijerph-20-01682-f001:**
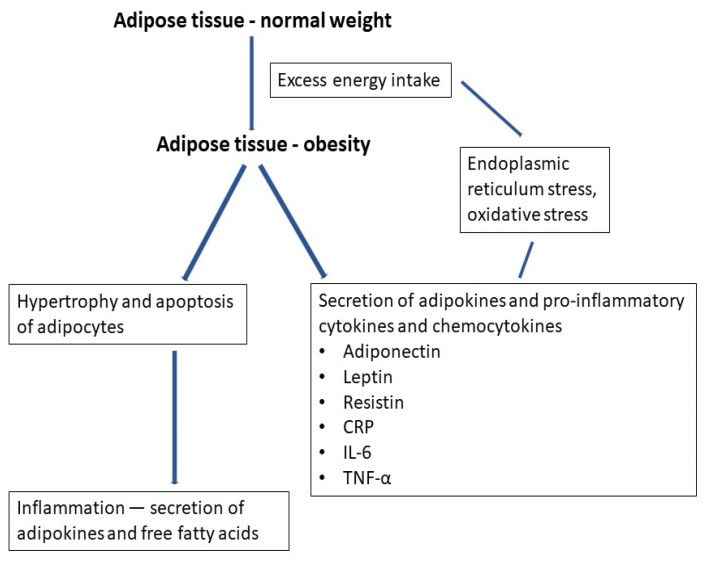
Mechanism of low-grade inflammation in obesity.

**Figure 2 ijerph-20-01682-f002:**
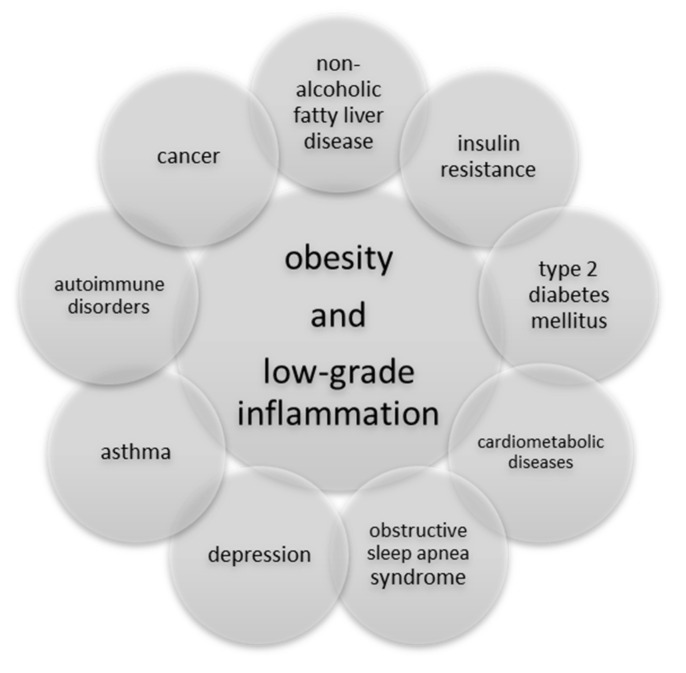
Possible consequences of low-grade inflammation in childhood obesity.

**Table 1 ijerph-20-01682-t001:** Pro-inflammatory and anti-inflammatory effect of diet in children.

	Normal Weight—Homeostasis	Overweight—Low-Grade Inflammation
Diet patterns	Mediterranean diet, Dietary Approaches to Stop Hypertension diet	Western-type diet
Foods groups	High vegetables, fruit, nuts, whole grains, legumes, cereals and roots intake	Ultra-processed food, sugar-sweetened beverages, high red meat, refined grains, high-fat dairy, sodium intake
Nutrients	High monounsaturated fatty acid [MUFA), omega-3 PUFA, vitamins A, E, C, β-carotene, luteolin, folate, fiber intake	Excess calories, high saturated or trans-fatty acids intake
Inflammatory markers	↓ IL-4, IL-6, IL-10, IL-17, TNF-α, hs-CRP	↑ IL-1, IL-2, IL-5, IL-6, T hs-CRP, NF-α, TGFβ-1, leptin, cellular adhesion molecule-1

↓—level decrease, ↑—level increase.

## Data Availability

Not applicable.

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
