# Peer review of "Low-Grade Inflammation and Role of Anti-Inflammatory Diet in Childhood Obesity"

_ijerph, 2023, doi:10.3390/ijerph20031682_

Round 1

Reviewer 1 Report (Previous Reviewer 1)

The authors can add a line or two about what sets apart childhood obesity from adult onset obesity to increase the importance of the manuscript.

All other queries have been answered satisfactorily

Author Response

Reviewer 2 Report (Previous Reviewer 2)

The author has made the suggested revisions and added further discussion as needed

Author Response

This manuscript is a resubmission of an earlier submission. The following is a list of the peer review reports and author responses from that submission.

Round 1

Reviewer 1 Report

In this manuscript, the authors have tried to describe the role of diets in the exacerbation of pediatric obesity.

1. What are the stats on pediatric obesity in the 2020s? Why have the authors stated data obtained in 2019 in the abstract?

2. The data needs to be segregated based on race, gender, and ethnicity. BMI is different for people of different ethnicity.

3. What diets are classified as anti-inflammatory and based on what? Are the studies listed here all specific to adolescents? How can diet be used for the treatment of pediatric obesity? What role does genetics play?

4. How is the mechanism described here specific to pediatric obesity?

5. The review is missing tables and figures. It is poorly written and is just an aggregation of information from here and there.

Author Response

RESPONSE TO REVIEWER 1 COMMENTS

I would like to express my appreciation to the reviewers and editorial board for taking the time and effort to improve our work and provide such insightful comments.

I am pleased to have been given the opportunity to revise our manuscript entitled “Low-grade inflammation and role of anti-inflammator diet in childhood obesity”.

I have carefully reviewed your comments. Below I explain how I revised the paper based on your comments and recommendations.

All responses to reviewer comments are in blue.

  1. What are the childhood obesity statistics in 2020? Why did the authors include data obtained in 2019 in the abstract?

Response: I would like to thank the Reviewer for this comments. The WHO organization provides information on the occurrence of overweight and obesity in children only from the years given in the introduction. My goal was to show a global problem over the years, which is why the WHO data seems to be the most relevant. The information from the publications most often refers to specific regions with is not concern of my review. Nevertheless, I added information related to the prevalence of obesity in several countries during the covid-19 pandemic in 2020 year.

  1. Data should be segregated by race, gender and ethnicity. BMI is different for people of different ethnic backgrounds.

Response: I am grateful for these comments. Adult BMI standards vary by ethnic group. On the other hand, the assessment of BMI in children is based on percentile charts for a specific age and sex. The WHO Growth Charts are taken into account in many countries. In my review, I used articles that took these conclusions into account when evaluating children's weight based on BMI. However, despite these considerations, I will again mention that the assessment of BMI was taken up in the research papers used by me. The aim of the article was not to consider the BMI index, but the relationship between obesity, low-grade inflammation and diet by analysing level of inflammation markers.

  1. What diets are considered anti-inflammatory and on what grounds? Are all studies listed here relevant to adolescents? How to use a diet to treat obesity in children? What role does genetics play?

Response: I would like to thank the Reviewer for this comments. The influence of genetics on obesity in children was not the subject of this article but I mentioned about genes in limitations. All studies considered related to diet and in the discussion concerned children, unfortunately their number is limited. An anti-inflammatory diet is a diet that has anti-inflammatory elements (food or nutrients). An anti-inflammatory diet can reduce level of inflammatory markers.

  1. How is the mechanism described here specific to childhood obesity?

Response: I would like to thank the Reviewer for this comment. It is suspected that the mechanism is the same as in adults, but it is not studied and described. An important problem is the lack of literature on this mechanism only in children, which I wrote about in the Methods and Limitations.

  1. The review lacks tables and figures. It is poorly written and is just an aggregation of information from here and there.

Response: Thank you very much for your valuable comments and efforts to improve my manuscript. I add table and figure. My goal was to create narrative review and no systematic search of the literature was performed.

Below is a certificate confirming the corrections made by a native speaker.

I did my best to improve the manuscript. I hope that the revised version will meet your expectations.

Thank you.

Reviewer 2 Report

This is a review of the data linking inflammation and metabolic disease in children with obesity. It covers mechanism and in human studies extensively cites research suggesting a correlation between obesity and elevated inflammatory cytokines. This is a mechanistic link that has been observed in animal models as well as human studies. The review also covers data on human trials of diets and their effect on these same markers.

The paper is well organized and provides references to support its findings. The English is understandable however grammatically there is need for some revision and editing. 

The mechanism section is somewhat dated. I think the field has expanded to include not just inflammation but also ER stress and oxidative stress, autophagy, and other mechanisms linking obesity and dysmetabolism. I also believe the dietary studies that are reviewed should provide more data about the trials - eligibility criteria, intervention, duration, and control for confounders. Changes in inflammatory markers may be due to changes in weight primarily or body composition, rather than specific to the diet itself. Also energy restriction could explain these changes rather than intrinsic properties of the diets.

Author Response

RESPONSE TO REVIEWER 2 COMMENTS

I would like to express my appreciation to the reviewers and editorial board for taking the time and effort to improve our work and provide such insightful comments.

I am pleased to have been given the opportunity to revise our manuscript entitled “Low-grade inflammation and role of anti-inflammatory diet in childhood obesity”.

I have carefully reviewed your comments. Below I explain how I revised the paper based on your comments and recommendations.

All responses to reviewer comments are in blue.

This is a review of the data linking inflammation and metabolic disease in children with obesity. It covers mechanism and in human studies extensively cites research suggesting a correlation between obesity and elevated inflammatory cytokines. This is a mechanistic link that has been observed in animal models as well as human studies. The review also covers data on human trials of diets and their effect on these same markers.

The paper is well organized and provides references to support its findings. The English is understandable however grammatically there is need for some revision and editing. 

 Response: I would like to thank the Reviewer for this comments. Below is a certificate confirming the corrections made by a native speaker.

The mechanism section is somewhat dated. I think the field has expanded to include not just inflammation but also ER stress and oxidative stress, autophagy, and other mechanisms linking obesity and dysmetabolism. I also believe the dietary studies that are reviewed should provide more data about the trials - eligibility criteria, intervention, duration, and control for confounders. Changes in inflammatory markers may be due to changes in weight primarily or body composition, rather than specific to the diet itself. Also energy restriction could explain these changes rather than intrinsic properties of the diets.

Response: Thank you very much for your valuable comments and efforts to improve my manuscript. As noted, I added a paragraph regarding the correlation of low-grade inflammation with ER stress, oxidative stress and autophagy. Unfortunately, this is such a vast topic that I still realize that I have not exhausted the topic. An important problem is the lack of literature on this mechanism only in children, which I wrote about in the Methods and Limitations.

Also in the Limitations I described the problem of changes in inflammatory markers that may result from changes in body weight or other factors, not from a change in diet.

I did my best to improve the manuscript. I hope that the revised version will meet your expectations.

Thank you.

Reviewer 3 Report

Please find my comments in the attached document.

Author Response

RESPONSE TO REVIEWER 3 COMMENTS

I would like to express my appreciation to the reviewers and editorial board for taking the time and effort to improve our work and provide such insightful comments.

I am pleased to have been given the opportunity to revise our manuscript entitled “Low-grade inflammation and role of anti-inflammatory diet in childhood obesity”.

I have carefully reviewed your comments. Below I explain how I revised the paper based on your comments and recommendations.

All responses to reviewer comments are in blue.

Thank You for the opportunity to review the manuscript titled: “Low-grade inflammation and role of anti-inflammatory diet in childhood obesity”.

The manuscript in question provides a critical evaluation of evidence on the potential role low grade inflammation and inflammation-regulatory diet in childhood obesity. This is an important contribution to the field as it presents a step forward to the understanding of the consequences of obesity in children.

In addition, the article is written in good English, thoughts and ideas are organized in a naturally flowing text making it an interesting and exciting read.

Response: I would like to thank the Reviewer for this comments. Below is a certificate confirming the corrections made by a native speaker.

However, there is a certain, minor issue that need to be resolved.

In the Materials and methods section, it should be made clear how the search was performed in terms of the language/s used.

It should be stated whether only articles written in English were included in the study, or if the study included data published in other languages.

If non-English articles were indeed used during the search and writing process, the potential cultural (language-related) bias should be properly addressed, possibly in the Limitations section (lines 389 and so on).

Response: Thank you very much for your valuable comments and efforts to improve my manuscript. I corrected the Methods section, highlighting that English-only articles were used. I also extended the information on how research was performed.

Line 49.

The sentence “This mechanism also mediated by the hormone leptin (5)” lacks something. Probably “is”, but this should be corrected.

Line 51.

“Aa” needs to be changed to “as”.

Lines 66-68.

The sentence “In children as young as 3 years of age there occurs a strong relation between waist circumference or body mass index and the level of CRP, IL-6 and tumour necrosis factor α (TNF-α) (8)” is almost exactly the same as the sentence in lines 72-74.

Lines 143-145

In the sentence “Children aged 14 with a higher BMI who were examined at the ages of 17 were more often suffered from excess body weight, inflammation, depression and other mental health problems (33).” – the syntax “were more often” should be just “more often” without “were”.

Line 174

“anty-inflammatory” is misspelled.

Line 192

The word “different” is misspelled.

Lines 203-204

The sentence states that “Research on adolescent show significant reduction of high-sensitivity C-reactive protein (52).” But it is unclear how this is linked to the previous sentence. Please elaborate on this issue. Is it the DASH diet in whole that reduces CRP? Is it the sodium level (as the reader might suggest based on the sequence of sentences)?

Response: I would like to thank the Reviewer for comments above. All have been corrected.

I did my best to improve the manuscript. I hope that the revised version will meet your expectations.

Thank you.